# DISTILLED EMBEDDING: NON-LINEAR EMBEDDING FACTORIZATION USING KNOWLEDGE DISTILLATION

## ABSTRACT

Word-embeddings are a vital component of Natural Language Processing (NLP) systems and have been extensively researched. Better representations of words have come at the cost of huge memory footprints, which has made deploying NLP models on edge-devices challenging due to memory limitations. Compressing embedding matrices without sacrificing model performance is essential for successful commercial edge deployment. In this paper, we propose Distilled Embedding, an (input/output) embedding compression method based on low-rank matrix decomposition with an added non-linearity. First, we initialize the weights of our decomposition by learning to reconstruct the full word-embedding and then fine-tune on the downstream task employing knowledge distillation on the factorized embedding. We conduct extensive experimentation with various compression rates on machine translation, using different data-sets with a shared word-embedding matrix for both embedding and vocabulary projection matrices. We show that the proposed technique outperforms conventional low-rank matrix factorization, and other recently proposed word-embedding matrix compression methods.

## 1 INTRODUCTION

Deep Learning models are the state-of-the-art in NLP, Computer Vision, Speech Recognition and many other fields in Computer Science and Engineering. The remarkable deep learning revolution has been built on top of massive amounts of data (both labeled and unlabeled), and faster computation. In NLP, large pre-trained language models like BERT (Devlin et al., 2018) are state-of-the-art on a large number of downstream NLP problems. The largest publicly available language model to date is trained with 1.6 billion parameters (Keskar et al., 2019). On machine translation the state-of-the-art models have parameters in the order of millions. Data privacy and server cost are some major issues, driving research towards deploying these models on edge-devices. However, running these models on edge-devices, face memory and latency issues due to limitations of the hardware. Thus, there has been considerable interest towards research in reducing the memory footprint and faster inference speed for these models (Sainath et al., 2013; Acharya et al., 2019; Shu & Nakayama, 2017; Shi & Yu, 2018; Jegou et al., 2010; Chen et al., 2018; Winata et al., 2018).

The architecture of deep-learning-based NLP models can be broken down into three components. The first component, represents the embedding section, which maps words in the vocabulary to continuous dense vector representations of the words. For all future references, the first component includes both the source and target vocabulary mapping using a common embedding matrix with a shared vocabulary. The second component, consists of a function $f$, typically a deep neural-network (Schmidhuber, 2015; Krizhevsky et al., 2012; Mikolov et al., 2010) which maps the embedding representation for different NLP problems (machine-translation, summarization, question-answering and others), to the output-space of function $f$. The third component, is the output layer which maps the output of function $f$ to the vocabulary-space, followed by a softmax function. Most of these NLP problems use linear transformations, in the first and the third components of the models. Since, these components depend upon a large vocabulary-size, they require large number of parameters which results in higher latency and larger memory requirements. For instance, the Transformer Base model (Vaswani et al., 2017) uses 37% of the parameters in the first and third components using a vocabulary size of 50k, and with parameter-tying between the components. The percentage of parameters increases to 54%, when parameters are not shared between the first and third components. The reader is encouraged to look at Table A.1 of the Appendix for more information.

Thus, an obvious step is to compress parametric functions used by the first and third components. Recently, many researchers have worked on compressing word-embedding matrices (Sainath et al., 2013; Acharya et al., 2019; Shu & Nakayama, 2017; Shi & Yu, 2018; Jegou et al., 2010; Chen et al., 2018; Winata et al., 2018). These techniques have proven to perform at-par with the uncompressed models, but still suffer from a number of issues.

**First**, embedding compression models (Shi & Yu, 2018; Chen et al., 2018; Khrulkov et al., 2019; Shu & Nakayama, 2017), require two hyper-parameters to be fine-tuned. These hyper-parameters influence the number of parameters in the model, and thus the compression rate. This leads to an additional layer of complexity for optimizing the model for different NLP problems. Additionally, Chen et al. (2018) requires an additional optimization step for grouping words, and lacks end-to-end training through back-propagation. Shi & Yu (2018) also requires an additional step for performing k-means clustering for generating the quantization matrix. Thus, most of the current state-of-the-art systems are much more complicated to fine-tune for different NLP problems and data-sets.

**Second**, all the state-of-the-art embedding compression models compress the first and third components separately. In practice, state-of-the-art NLP models like Vaswani et al. (2017) have shown to perform better with parameter sharing between the first and third components (Press & Wolf, 2016). Thus, there is a need for an exhaustive analysis of various embedding compression techniques, with parameter sharing.

**Lastly**, embedding compression models not based on linear SVD (Khrulkov et al. (2019); Shu & Nakayama (2017); Shi & Yu (2018)) require the reconstruction of the entire embedding matrix or additional computations, when used at the output-layer. Thus, the model either uses the same amount of memory as the uncompressed model or requires additional computation cost. This makes linear SVD based techniques more desirable for running models on edge-devices.

In this paper, we introduce Distilled Embedding, a non-linear matrix factorization method, which outperforms the current state-of-the-art methods. Our method, first compresses the vocabulary-space to a much smaller dimension compared to the original hidden-dimension, then applies a non-linear activation function, before recovering the original embedding-dimension. Additionally, we also introduce an embedding distillation method, which is similar to Knowledge Distillation (Hinton et al., 2015) but we apply it to distill knowledge from a pre-trained embedding matrix and use an $L2$ loss instead of cross-entropy loss. To summarize our contributions are:

- We demonstrate that at the same compression rate our method outperforms existing state-of-the-art methods.
- Our proposed method is much simpler than the current state-of-the-methods, with only a single hyper-parameter controlling the compression rate.
- We introduce an embedding distillation method, which out-performs standard machine translation cross-entropy loss.
- Unlike the current state-of-the-art systems, we compress the embedding matrix with parameter sharing between the first and third components. We perform an exhaustive comparison of current state-of-the-art models in this setting.
- Our model uses a similar factorization as SVD, with the addition of a non-linearity, thus it is able to leverage benefits of linear SVD based methods discussed above.
- We also compare all the models against a simple low-rank matrix factorization using Singular Value Decomposition (SVD) and demonstrate that it can compete with more complex algorithms.

## 2   RELATED WORK

We can model the problem of compressing the embedding matrix as a matrix factorization problem. There is a considerable amount of work done in this field, some of the popular works broadly belong to the domain of low-rank factorization (Singular Value Decomposition (SVD);Srebro & Jaakkola (2003); Mnih & Salakhutdinov (2008), product quantization (Jegou et al., 2010) and tensor decomposition (De Lathauwer et al., 2000). A number of prior works in embedding compression are influenced by these fields and have been applied to various NLP problems. In this Section, we will discuss some of the significant works across different NLP problems.

**Low-rank Factorization**   Low-rank approximation of weight matrices, using SVD, is a natural way to compress deep learning based NLP models. Sainath et al. (2013) apply this to a convolutional neural network for language modeling and acoustic modeling. Winata et al. (2018) use SVD on all the weight matrices of an LSTM and demonstrate competitive results on question-answering, language modeling and text-entailment. Acharya et al. (2019) use low-rank matrix factorization for word-embedding layer during training to compress a classification model. However, they do not study the effects of applying a non-linear function before reconstructing the original dimension.

**GroupReduce**   Chen et al. (2018) apply weighted low-rank approximation to the embedding matrix of an LSTM. They first create a many-to-one mapping of all the words in the vocabulary into $g$ groups, this initial mapping is done based upon frequency. For each group $g$ they apply weighted SVD to obtain a lower rank estimation, the rank is determined by setting the minimum rank and linearly increasing it based upon average frequency. Finally, they update the groups by minimizing the reconstruction error from the weighted SVD approximation. They demonstrate strong results on language modeling and machine translation compared to simple SVD. In their models they use different embedding matrices for input and softmax layers and apply different compression to each.

**Product Quantization**   Jegou et al. (2010) introduced product quantization for compressing high dimensional vectors, by uniformly partitioning them into subvectors and quantizing each subvector using K-means clustering technique. Basically, product quantization approach assumes that the subvectors share some underlying properties which can be used to group similar subvectors together and unify their representation. That being said, this approach breaks the original matrix into a set of codebooks coming from the center of the clusters in different partitions together with a separate index matrix which refers to the index of the clusters for each subvector. Shi & Yu (2018) applied product quantization to a language model and were able to show better perplexity scores. Shu & Nakayama (2017) extended this technique by first representing the product quantization as a matrix factorization problem, and then learning the quantization matrix in an end-to-end trainable neural network. Li et al. (2018) implement product quantization through randomly sharing parameters in the embedding matrix, and show good results on perplexity for an LSTM based language model.

**Tensor Decomposition**   De Lathauwer et al. (2000) introduced multilinear SVD, which is a generalization of SVD for higher order tensors. Oseledets (2011) introduced an efficient algorithm Tensor Train (TT) for multilinear SVD Tensor. Novikov et al. (2015) applied the Tensor Train decomposition on fully connected layers of deep neural networks. Khrulkov et al. (2019) applied Tensor Train algorithm to the input embedding layer on different NLP problems like language modeling, machine translation and sentiment analysis. They demonstrate high compression rate with little loss of performance. However, they compress only the input embedding and not the softmax layer for language modeling and machine translation.

**Knowledge Distillation**   Knowledge distillation has been studied in model compression where knowledge of a large cumbersome model is transferred to a small model for easy deployment. Several studies have been studied on the knowledge transfer technique (Hinton et al. (2015); Romero et al. (2015)). In this paper, we propose a embedding factorization of word-embedding matrix using knowledge distillation to mimic the pre-trained word-embedding representation.

## 3   METHODOLOGY: DISTILLED EMBEDDING

### 3.1   FUNNELING DECOMPOSITION AND EMBEDDING DISTILLATION

We present an overview of our proposed method in Figure 1. Given an embedding matrix $E \in \mathbb{R}^{|\mathcal{V}| \times d}$, we can decompose it into three matrices (Equation 1), using the SVD algorithm

$$E = U_{|\mathcal{V}| \times |\mathcal{V}|} \Sigma_{|\mathcal{V}| \times d} V_{d \times d}^T \tag{1}$$

where $|\mathcal{V}|$ is the vocabulary size and $d$ is the embedding dimension. $\Sigma$ is a diagonal matrix containing the singular values, and matrices $U$ and $V$ represent the left and right singular vectors of the embedding matrix respectively. We can obtain the reduced form of the embedding matrix by only keeping $r$ ($< d$) largest singular values out of $d$.

$$\tilde{E} = U_{|\mathcal{V}| \times r} \Sigma_{r \times r} V_{r \times d}^T = \mathbb{U}_{|\mathcal{V}| \times r} V_{r \times d}^T \tag{2}$$

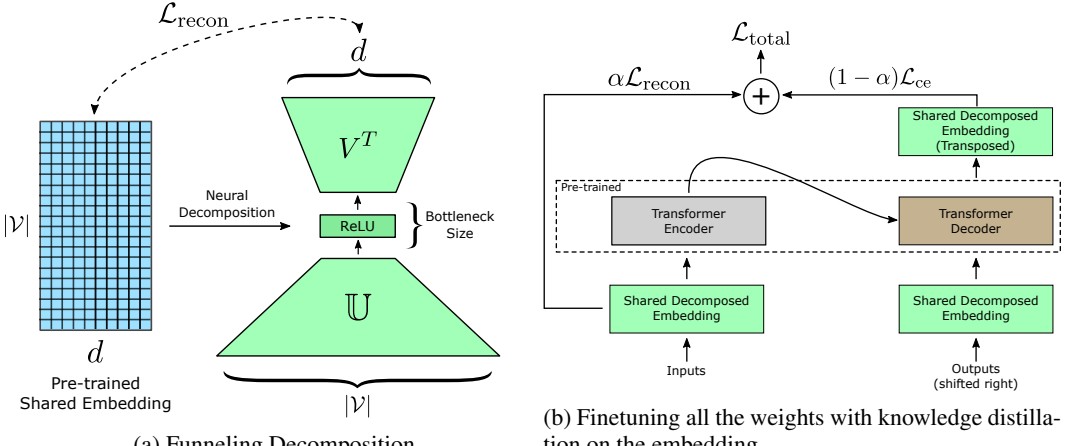

Figure 1: Funneling Decomposition method to compress the shared embedding matrix of a transformer based sequence to sequence model.

where the matrix $\mathbb{U} = U\Sigma$. The reduced form of the embedding matrix will need $r \times (|\mathcal{V}| + d)$ parameters compared to $|\mathcal{V}| \times d$.

Our proposed approach in this work, is to apply a non-linear transformation on the matrix $\mathbb{U}$, before reconstructing the original embedding dimension using $V$ (see Figure 1a), as shown in Equation 3,

$$\tilde{E} = f(\mathbb{U}_{|\mathcal{V}| \times r})V_{r \times d}^T \tag{3}$$

We use the ReLU as our non-linear function $f(.)$ throughout this paper. We train a sequence to sequence model (Sutskever et al. (2014); Vaswani et al. (2017)) with tied input and output embedding (i.e. the output embedding is the transpose of the input embedding matrix $\tilde{E}_{\text{out}} = \tilde{E}^T = V_{d \times r}[f(\mathbb{U}_{|\mathcal{V}| \times r}]^T$. We train our model end-to-end by replacing the embedding function with Equation 3. The matrix $\mathbb{U}$ and $V$ are trainable parameters, and for the output layer we use $\tilde{E}^T$, with the parameter sharing. We train on two losses. The standard cross entropy loss defined as:

$$\mathcal{L}_{\text{ce}} = -\sum_{i=1}^{M} \mathrm{y}_i \log(p_i) \tag{4}$$

where $M$ is the sequence length, $\mathrm{y}_i$ is the one-hot representation for the $i^{\text{th}}$ label and $p_i$ is the softmax probability of the $i^{\text{th}}$ term generated by the decoder.

In addition to the cross-entropy loss, we introduce a novel embedding reconstruction loss (Equation 5), which we refer to as embedding distillation as we distill information from the pre-trained embedding into our model,

$$\mathcal{L}_{\text{recon}} = \frac{1}{|\mathcal{V}|} \sum_{i=1}^{|\mathcal{V}|} \|e_i - \hat{e}_i\|_2 = \frac{1}{|\mathcal{V}|} \sum_{i=1}^{|\mathcal{V}|} \|e_i - f(u_i)V_{r \times d}^T\|_2 \tag{5}$$

where $e_i$ and $\hat{e}_i$ are the embedding vectors corresponding to the $i^{\text{th}}$ word in the original embedding matrix $E$ and the reconstructed embedding matrix $\hat{E}$ respectively and $u_i$ refers to the $i^{\text{th}}$ row of the matrix $\mathbb{U}$. We use Equation 6 as our final loss function.

$$\mathcal{L}_{\text{total}} = \alpha \mathcal{L}_{\text{recon}} + (1 - \alpha)\mathcal{L}_{\text{ce}} \tag{6}$$

where $\alpha \in [0, 1]$ is a hyper-parameter, which controls the trade-off between reconstruction and cross-entropy loss. $\mathcal{L}_{\text{recon}}$ acts as the knowledge distillation loss by which we try to distill information from the original pre-trained embedding layer as a teacher to the funneling decomposed embedding layer as a student. The training process of our Distilled Embedding method is summarized in **Algorithm 1**.

---

**Algorithm 1** Distilled Embedding

---

**Step 1) Pre-training the Embedding Matrix** Pre-train the sequence to sequence model with the full embedding matrix for better initialization.

**Step 2) Initializing the Weights of Funneling Decomposition Layer** We extract the trained embedding matrix $E$ from Step 1 and train our decomposed matrices $\mathbb{U}$ and $V$ on reconstruction loss defined in Equation 5, as shown in Figure 1a.

**Step 3) Embedding Distillation** The pre-trained funneling decomposition layer is plugged into the model (replacing the original embedding matrix $E$) and the entire model is trained based on Equation 6.

---

## 4 EXPERIMENTAL SETUP

### 4.1 DATASETS AND EVALUATION

We test our proposed method on machine translation which is a fundamental problem in NLP and challenging for embedding compression since we typically have at least two dictionaries and an input and output embedding. Comparitively language modeling uses a single dictionary and classification tasks such as sentiment analysis don't have an output embedding.

We present results on translating three language pairs: WMT English to French (En-Fr), WMT English to German (En-De) and IWSLT Portuguese to English (Pt-En). We decided that these pairs are good representatives of high-resource, medium-resource and low-resource language pairs.

WMT En-Fr is based on WMT14 training data which contain 36M sentence pairs. We used SentencePiece (Kudo & Richardson (2018)) to extract a shared vocabulary of 32k subwords. We validate on newstest2013 and test on newstest2014. For WMT English to German (En-De), we use the same setup as Vaswani et al. (2017). The dataset is based on WMT16 training data and contains about 4.5M pairs. We use a shared vocabulary of 37k subwords extracted using SentencePiece.

For the IWSLT Portuguese to English (Pt-En) dataset, we replicate the setup of Tan et al. (2019) for training individual models. Specifically, the dataset contains about 167k training pairs. We used a shared vocabulary of 32k subwords extracted with SentencePiece.

For all language pairs, we measure case-sensitive BLEU score (Papineni et al. (2002)) using Sacre-BLEU[1] (Post (2018)). In addition, we save a checkpoint every hour for the WMT En-Fr and WMT En-De language pairs and every 5 minutes for the IWSLT Pt-En due to the smaller size of the dataset. We use the last checkpoint which resulted in the highest validation BLEU and average the last five checkpoints based on this. We use beam search with a beam width of 4 for all language pairs.

### 4.2 EXPERIMENT DETAILS

**Hyper-Parameters** For WMT En-Fr and WMT En-De we use the same configuration as Transformer Base which was proposed by Vaswani et al. (2017). Specifically, the model hidden size $d_{\text{model}}$ is set to 512, the feed-forward hidden size $d_{\text{ff}}$ is set to 2048 and the number of layers for the encoder and the decoder was set to 6. For the IWSLT Pt-En, we use Transformer Small configuration. Specifically, the model hidden-size $d_{\text{model}}$ is set to 256, the feed-forward hidden size $d_{\text{ff}}$ is set to 1024 and the number of layers for the encoder and the decoder was set to 2. For Transformer Small, the dropout configuration was set the same as Transformer Base. All models are optimized using Adam (Kingma & Ba (2015)) and the same learning rate schedule as proposed by Vaswani et al. (2017). We use label smoothing with 0.1 weight for the uniform prior distribution over the vocabulary (Szegedy et al. (2015); Pereyra et al. (2017)). Additionally, we set the value $\alpha$ of Equation 6 to 0.01.

**Hardware Details** We train the WMT models on 8 NVIDIA V100 GPUs and the IWSLT models on a single NVIDIA V100 GPU. Each training batch contained a set of sentence pairs containing

---

[1]`https://github.com/mjpost/sacreBLEU`

| Model | Param | Emb. Params | Emb. Compression Rate | WMT En-Fr (BLEU) |
|---|---|---|---|---|
| Transformer Base | 60M | 16.3M | 1.0x | 38.12 |
| Smaller Transformer Network (416) | 46M | 13.3M | 1.23x | 37.26 |
| SVD with rank 64 | 46M | 2.08M | 7.87x | 37.44 |
| End-to-End NN with non-linearity | 46M | 2.08M | 7.87x | 37.23 |
| GroupReduce (Chen et al. (2018)) | 46M | 2.10M | 7.79x | 37.63 |
| Structured Embedding (Shi & Yu (2018)) | 46M | 2.07M | 7.90x | **37.78** |
| Tensor Train (Khrulkov et al. (2019)) | 46M | 2.12M | 7.72x | 37.27 |
| Distilled Embedding (Ours) | 46M | 2.08M | 7.87x | **37.78** |

Table 1: Machine translation accuracy in terms of BLEU for WMT En-Fr on newstest2014.

approximately 6000 source tokens and 6000 target tokens for each GPU worker. All experiments were run using the TensorFlow framework[2].

## 5 RESULTS

### 5.1 MACHINE TRANSLATION

We present BLEU score for our method and compare it with SVD, GroupReduce (Chen et al. (2018)), Structured Emedding (Shi & Yu (2018)), Tensor Train (Khrulkov et al. (2019)) and a smaller transformer network with the same number of parameters. We learn a decomposition for all the methods except Tensor Train since it was pointed out in Khrulkov et al. (2019) that there is no difference in performance between random initialization and tensor train learnt initialization. Once initialized we plug the decomposed embedding and fine-tune till convergence. None of the weights are frozen during fine-tuning.

Table 1 presents the results on English-French translation. The hyper-parameters for tuning the competing methods are presented in Section A.1 of the Appendix. We see that on this task our method along with Structured Embedding performs the best. Group Reduce is the next, and SVD performs better than Tensor Train, showing that SVD is a strong baseline, when fine-tuned till convergence. We also compare against a 2 layer neural network (NN) with the same parameterization as distilled embedding which has not been initialized offline. The results show that initializing the NN with the embedding weights is important.

On English-German translation, as seen in Table 2, our method outperforms all other methods. The smaller transformer network does well and is only surpassed by GroupReduce amongst the competing methods. SVD again performs better than Tensor Train.

We present the Portuguese-English translation results on Table 3. This task presents a problem where the embedding matrix constitutes the majority of the parameters of the neural network. The embedding dimension is smaller (256) compared to the other two tasks but embedding compression yields a BLEU score increase in all methods except Structured Embedding. This is due to a regularization effect from the compression. Our model again achieves the highest BLEU score.

On these three experiments we demonstrate that our funneling decomposition method with embedding distillation consistently yields higher BLEU scores compared to the competing methods.

### 5.2 ABLATION STUDY

We present different experiments to demonstrate the effect of 1) Model Initialization, 2) Embedding Distillation, 3) Fine-tuning strategies, 4) Compression capability and 5) Extension and generality of our method.

---

[2]https://www.tensorflow.org/

| Model | Param | Emb. Params | Emb. Compression Rate | WMT En-De (BLEU) |
|---|---|---|---|---|
| Transformer Base | 63M | 18.94M | 1.0x | 27.08 |
| Smaller Transformer Network (400) | 46M | 14.8M | 1.28x | 26.72 |
| SVD with rank 64 | 46M | 2.40M | 7.89x | 26.32 |
| End-to-End NN with non-linearity | 46M | 2.40M | 7.89 | 26.14 |
| GroupReduce (Chen et al. (2018)) | 46M | 2.40M | 7.88x | 26.75 |
| Structured Embedding (Shi & Yu (2018)) | 46M | 2.40M | 7.89x | 26.34 |
| Tensor Train (Khrulkov et al. (2019)) | 46M | 2.44M | 7.75x | 26.19 |
| Distilled Embedding (Ours) | 46M | 2.40M | 7.89x | **26.97** |

Table 2: Machine translation accuracy in terms of BLEU for WMT En-De on newstest2014.

| Model | Param | Emb. Params | Emb. Compression Rate | IWSLT Pt-En (BLEU) |
|---|---|---|---|---|
| Transformer Small | 11M | 8.19M | 1.0x | 41.43 |
| Smaller Transformer Network (136) | 5M | 4.35M | 1.88x | 40.71 |
| SVD with rank 64 | 5M | 2.06M | 3.96x | 42.37 |
| End-to-End NN with non-linearity | 5M | 2.06M | 3.96 | 42.27 |
| GroupReduce (Chen et al. (2018)) | 5M | 2.06M | 3.96x | 42.13 |
| Structured Embedding (Shi & Yu (2018)) | 5M | 2.06M | 3.97x | 41.27 |
| Tensor Train (Khrulkov et al. (2019)) | 5M | 2.06M | 3.96x | 42.34 |
| Distilled Embedding (Ours) | 5M | 2.06M | 3.96x | **42.62** |

Table 3: Machine translation accuracy in terms of BLEU for IWSLT Pt-En.

**Initialization**   We do an ablation study on all the three language pairs defined in Section 4.1, to conclude, if random initialization is better than model based initialization. We conclude that model based initialization, consistently performs better (Table 4).

**Embedding Distillation**   Table 5 presents different compression rates on the Pt-En task, and embedding distillation performs better across all of them. In Table 4, we see that across all language pairs when we initialize our model using weights from the funneling decomposition, we improve when using Embedding Distillation during finetuning. We performed embedding distillation with random initialization only on the smaller Pt-En dataset and observed that Embedding Distillation improves BLEU score.

**Compression Rate**   We demonstrate in Table 5 that it is possible to compress the embedding up to 15.86x with only a 2% drop in BLEU score for Pt-En.

**Re-training**   Fine-tuning is an important component in our method and we demonstrate through our experiments that at convergence most of the techniques are close in performance. Table 6 shows that freezing embedding weights and re-training the network weights or vice versa leads to a sharp

| Model | Params | Emb. Params | Emb. CR | Random Initialization | | Model Initialization | |
|---|---|---|---|---|---|---|---|
| | | | | No Distillation | Emb. Distillation | No Distillation | Emb. Distillation |
| **En-Fr** | 46M | 2M | 7.87x | 37.04 | - | 37.54 | **37.78** |
| **En-De** | 46M | 2M | 7.89x | 26.07 | - | 26.7 | **26.97** |
| **Pt-En** | 5M | 2M | 3.96x | 42.29 | 42.36 | 42.5 | **42.62** |

Table 4: Comparison of different methods for Funneling (64), CR refers to the compression rate.

| Params | Emb. Params | Emb. CR | No Distillation | Emb. Distillation |
|--------|-------------|---------|-----------------|-------------------|
| 11M | 8M | 1.0x | **41.43** | - |
| 5M | 2M | 3.96x | 42.50 | **42.62** |
| 4M | 1M | 7.93x | 42.44 | **42.60** |
| 4M | 516k | 15.86x | 40.42 | **40.60** |

Table 5: Comparison of different compression rates with bottleneck sizes of 64, 32 and 16 accordingly for IWSLT Pt-En.

| Model | BLEU |
|-------|------|
| Proposal | **42.60** |
| - embedding distillation | 42.44 |
| - non-linearity | 42.34 |
| Proposal (Freeze non-emb. weights) | 33.34 |
| Proposal (Freeze emb. weights) | 20.49 |

Table 6: BLEU score for IWSLT Pt-En with compression rate 7.93x.

| Model | BLEU |
|-------|------|
| GroupFunneling (Rand. Initialized + Emb. Distil.) | **42.52** |
| GroupFunneling (Rand. Initialized) | 42.49 |
| GroupReduce | 42.13 |

Table 7: GroupFunneling (i.e. GroupReduce + Funneling) on IWSLT Pt-En.

drop in BLEU score, thus, we need to re-train all the weights. The use of a non-linearity and adding embedding distillation also improves BLEU score after finetuning.

**Extension**   We experimented with applying two key lessons from our method, namely, using a non-linear function and embedding distillation, to a model initialized with group partitions of the GroupReduce method (Chen et al., 2018), we refer to this method as *GroupFunneling*. Table 7 shows that, *GroupFunneling* achieves a higher BLEU score on Pt-En compared to GroupReduce.

## 6 DISCUSSION

**Importance of Non-linearity**   We postulate that only a subset of word vector dimensions, explains most of the variance, for most word vectors in the embedding matrix. Thus, using ReLU activation might help in regularizing the less important dimensions for a given word vector.

**Importance Reconstruction Loss**   We propose that the embedding reconstruction might suffer from adding the ReLU activation function. Thus, adding a loss for embedding reconstruction helps in grounding the embedding and not loose a lot of information. Thus, the amount of regularization is controlled by the hyper-parameter $\alpha$. Our intuition is partly justified by results shown in Table A.2, as reconstruction loss performs worse without the ReLU activation function.

## 7 CONCLUSION AND FUTURE WORK

In this paper we proposed Distilled Embedding, a low-rank matrix decomposition with non-linearity in the bottleneck layer for a shared word-embedding and vocabulary projection matrix. We also introduce knowledge distillation of the embedding during fine-tuning using the full embedding matrix as the teacher and the decomposed embedding as the student. We compared our proposed approach with state-of-the-art methods for compressing word-embedding matrix. We did extensive experiments using three different sizes of datasets and showed that our approach outperforms the state-of-the art methods on the challenging task of machine translation. For future work, we will apply our approach to compress feed-forward and multi-head attention layers of the transformer network.

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

# A APPENDIX

## A.1 ADDITIONAL HYPER-PARAMETERS

**WMT En-Fr** Smaller Transformer Network denotes a network with the same configuration as Transformer Base but with hidden size $d_{\text{model}}$ of 416. For GroupReduce, to match the same compression rate we used number of clusters $c$ being equal to 10 and minimum rank $r_{\text{min}}$ to be 22. For SVD, we decided to set the rank to 64. For Tensor Train, we set the embedding shape to be $[25, 32, 40] \times [8, 8, 8]$ and the Tensor Train Rank to be 90. For structured embedding we use group size as 32 and number of clusters as 2048, we then use the quantization matrix and learn the clusters from scratch.

**WMT En-De** Smaller Transformer Network denotes a network with the same configuration as Transformer Base but with hidden size $d_{\text{model}}$ of 400. For GroupReduce, to match the same compression rate we used number of clusters $c$ being equal to 10 and minimum rank $r_{\text{min}}$ to be 23. For SVD, we decided to set the rank to 64. For Tensor Train, we set the embedding shape to be $[25, 37, 40] \times [8, 8, 8]$ and the Tensor Train Rank to be 90. For structured embedding we use group size as 32 and number of clusters as 2376, we then use the quantization matrix and learn the clusters from scratch.

**IWSLT Pt-En** Smaller Transformer Network denotes a network with the same configuration as Transformer Small but with hidden size $d_{\text{model}}$ of 136. For GroupReduce, to match the same compression rate we used number of clusters $c$ being equal to 15 and minimum rank $r_{\text{min}}$ to be 30. For SVD, we decided to set the rank to 64. For Tensor Train, we set the embedding shape to be $[25, 32, 40] \times [8, 4, 8]$ and the Tensor Train Rank to be 125. For structured embedding we use group size as 32 and number of clusters as 4048, we then use the quantization matrix and learn the clusters from scratch.

## A.2 PARAMETER COUNT

Table A.1 presents the the number of parameters in the different transfomer layers for the transformer base architecture.

| Parameters | Embedding | FFN | Multi-head attention | Linear |
|---|---|---|---|---|
| Number | 26M | 25M | 14M | 5M |
| Percentage | 37% | 36% | 20% | 7% |

Table A.1: Parameters in the Transformer Base model (Vaswani et al. (2017)) based on a 50k dictionary size and tied input and output embedding.

## A.3 SVD WITH RECONSTRUCTION LOSS

Table A.2 presents BLEU scores for SVD on English French translation when combining reconstruction loss with SVD. This result suggests that SVD does not perform well with reconstruction loss.

| Model | BLEU |
|---|---|
| SVD with rank 64 | 37.44 |
| SVD with rank 64 with Recon. Loss (alpha 0.1) | 37.29 |

Table A.2: Comparison of SVD with and without reconstruction loss on En-Fr translation.

## A.4 CO-RELATION OF RECONSTRUCTION LOSS AND TRANSLATION

We present the L2 reconstruction losses for factorization of embedding matrices in table A.3. We see that a lower reconstruction loss does not necessarily lead to better translation performance. e.g.

GroupReduce has the lowest reconstruction loss on Pt-En but Funneling based solution gets higher BLEU score.

| Model | En-Fr Recon. Loss | En-De Recon. Loss | Pt-En Recon. Loss |
|---|---|---|---|
| SVD | 2.247 | 2.447 | 1.199 |
| GroupReduce | 2.247 | 2.445 | 1.173 |
| Funneling | 2.238 | 2.441 | 1.194 |

Table A.3: Reconstruction losses for all language pairs.

## A.5 HYPER-PARAMETER SENSITIVITY

We present the sensitivity of our algorithm to the hyper-parameter $\alpha$ for Pt-En translation in table A.4. We observe that the algorithm works well over a wide range of $\alpha$ values. In this work we did not tune $\alpha$ for different datasets but doing so might yield higher performance as is the case for Pt-En translation with $\alpha = 0.5$.

| $\alpha$ | BLEU |
|---|---|
| 0 | 42.5 |
| 0.01 | 42.62 |
| 0.1 | 42.65 |
| 0.3 | 42.66 |
| 0.5 | 42.72 |
| 0.7 | 42.57 |
| 0.9 | 42.03 |

Table A.4: BLEU score for Pt-En translation for different alpha values.

