# OpenReview forum: "Distilled embedding: non-linear embedding factorization using knowledge distillation"
_ICLR.cc/2020/Conference — Reject_

### Official Review · AnonReviewer2 · 2019-10-24
**Official Blind Review #2**

**Rating:** 3

**Review:**



This paper proposes a method for compressing embedding matrices of both encoder/decoder embeddings.
The basic idea of the proposed method is to reconstruct the embedding matrix by what they called the “funneling decomposition” method, whose parameter shape is identical to the SVD (low-rank matrix) decomposition with additional non-linear function.
Therefore, the idea itself is not so novel and innovative.
Moreover, their method requires the embedding matrix as the teacher signal for calculating the reconstruction loss.
We need to note that the memory requirement of the proposed method during training will increase.

One of the notable advantages of the proposed method is that their proposed method seems to successfully reduce the embedding matrix even if it shares the parameters with the output layer, which is a de-facto standard model architecture for NMT.
As pointed out by the authors, this seems to be the first success of reducing the embedding matrix with a tied embedding setting.


1,
The authors claim that “We demonstrate that at the same compression rate our method outperforms existing state-of-the-art methods.” at the end of the Introduction section.
However, according to Tables 1, 2, and 3, it seems that the performance gain is marginal compared with similar methods.
For example,
37.78 (proposed) <=> 37.78 (Shi & Yu (2018))
26.97 (proposed) <=> 26.75 (Chen et al. (2018)
and
 42.62 (proposed) <=> 42.37 (SVD with rank 64),
which are the at most 0.25 BLEU gain.
I believe that most of MT researchers hardly say that BLEU 0.25 difference is a significant improvement. Besides, the authors should perform a statistically significant test if they say “our method outperforms existing state-of-the-art methods.”


2
I am a bit confused about the following inconsistency;
The authors say that “Compressing embedding matrices without sacrificing model performance is essential for successful commercial edge deployment” in the abstract.
However, according to Table 1, the number of parameters for embeddings is 16.3M, which is only 27% of the total number of parameters in Transformer base.
By this fact, compressing embedding matrices seems not essential for successful commercial edge deployment.

In Table 6, it is explicitly unclear what is the difference between
“Funneling with Emb. Distillation”, “Funneling (with non-linearity),” and “Funneling (with retraining all weights).”
Please give us a more precise explanation.


**Experience Assessment:**

I have published in this field for several years.

**Review Assessment: Checking Correctness Of Derivations And Theory:**

I carefully checked the derivations and theory.

**Review Assessment: Checking Correctness Of Experiments:**

I carefully checked the experiments.

**Review Assessment: Thoroughness In Paper Reading:**

I read the paper thoroughly.

---

> ### Author Response · Authors · 2019-11-13
> **Response to Review#2**
>
> 1) We were careful not to use the word 'significantly' since we did not include any statistical significance tests. We agree that it gives a better account of the improvement in performance but for these models, the computational cost for these analyses would be prohibitive. We propose another way to compare the results of the proposed method against competing methods.
>
> Proposed vs Shi & Yu (2018):
>
> En-Fr       37.78 (proposed) <=> 37.78
> En-De      26.97 (proposed) <=> 26.34
> Pt-En       42.62 (proposed) <=> 41.27
>
>
> Proposed vs Chen et al. (2018):
>
> En-Fr       37.78 (proposed) <=> 37.63
> En-De      26.97 (proposed) <=> 26.75
> Pt-En       42.62 (proposed) <=> 42.13
>
>
> Proposed vs SVD rank 64:
>
> En-Fr       37.78 (proposed) <=> 37.44
> En-De      26.97 (proposed) <=> 26.32
> Pt-En       42.62 (proposed) <=> 42.37
>
> Based on this we conclude that we are consistently better and .49 BLEU better on at least one dataset. Our experimental philosophy was to use widely reported translation datasets, standard architectures and to re-train the models to convergence. This meant that the performance of all competing methods was closer than previously anticipated and our proposed method scored consistently higher BLEU scores compared to the rest.
>
> 2) We think that any commercial edge deployment of NLP models will combine a range of solutions including but not limited to weight quantization (depending on hardware), embedding compression, network weight reduction, parameter sharing and knowledge distillation. Embedding matrices, in the experiments we presented, constitute  27% (En-Fr) to 74.45% (Pt-En) of the network parameters. So depending on the rest of the model, embedding compression may help us shave 23% to 63% of the model (with our solution) without much loss of performance. Any solution, other than quantization, which aims to reduce model size would need to compress or reduce the embedding size. Quantization does not reduce the number of parameters but reduces the storage size. However, it is hardware dependant and not always a viable option.
>
> Moreover, embedding matrices are present in all NLP applications and constitute a majority of parameters for smaller models.
>
> Regarding Table 6 we thank you for noticing. We updated the formatting to make it clear. We have revised the submission but briefly:
>
> Model                                                         BLEU
> Proposal                                                     42.60
>     - embedding dist.                                 42.44
>     - non-linearity                                       42.34
> Proposal (Freeze non-emb weights)     33.34
> Proposal (Freeze emb. weights)            20.49
>
> We compare the proposal against removing embedding distillation and removing embedding distillation and non-linearity. We also show the effect of freezing the embedding weights during fine-tuning and freezing the non-embedding weights.

---

### Official Review · AnonReviewer1 · 2019-10-24
**Official Blind Review #1**

**Rating:** 3

**Review:**

The paper proposes to use low-rank matrix decomposition for embedding compression, with relu in the reconstruction layer to gain non-linearity. Experiments on machine translation task shows improvement compared with state-of-the-art methods with different compression rates.

Detailed comments:
1)	The technical contribution seems to be a bit limited. Using relu in the reconstruction function looks straightforward and adding reconstruction loss in objective function is also common practice. Also, not much insight is provided on why such approach works better than other baselines.

2)	Experiments:
a.	It is good to see such simple approach outperforms several more sophisticated baseline methods. Also, ablation study is also performed to show the effect of different components.

b.	How does the time complexity and running time of the proposed method compared to the baselines?

c.	The paper only evaluates distilled embedding on one task (i.e., machine translation). The experiments would be more convincing if evaluated on more tasks as well.

d.	It could be helpful to include some sensitivity analysis on the hyperparameters such as \alpha which controls the weight of reconstruction loss.

In conclusion, this paper seems to be below the bar and I would recommend a ‘weak reject’ for the paper.


**Experience Assessment:**

I have read many papers in this area.

**Review Assessment: Checking Correctness Of Derivations And Theory:**

I assessed the sensibility of the derivations and theory.

**Review Assessment: Checking Correctness Of Experiments:**

I assessed the sensibility of the experiments.

**Review Assessment: Thoroughness In Paper Reading:**

I read the paper at least twice and used my best judgement in assessing the paper.

---

> ### Author Response · Authors · 2019-11-14
> **Response to Review #1**
>
> Thank you for your review and critique.
>
> 1) The main strengths of our model are simplicity, a single parameter to control compression rate (bottleneck size), no reliance on frequency information which is seldom available for pre-trained models and a faster running time which we demonstrate in Section 2 b). Although in hindsight the approach looks simple, which we think is a strength, we considered many factors and ran many experiments to come up with this approach. The embedding reconstruction loss was an innovation that we found consistently performed better and we did not come across any embedding reduction/compression paper which tried it. However, in other works we agree that weight reconstruction loss is common.
>
> 2)
>
> b) We are running an analysis on that and will update our response with it.
>
> c) We ran an experiment on language modelling using the transformer-xL on wiki-text 103. We replicated the setup on the Github repository [1] and were able to achieve a similar test perplexity. We then tried SVD and Distilled Embedding (proposal) with a bottleneck of 32 to compress the model 12.79x times. We also present the result on Distilled Embedding at 6.32x compression with a bottleneck size of 64.
>
> Wikitext-103:
>
> | Model                                   | Compression  | Val PPl | Test PLL |
> |                                               |                           |              |                 |
> | Transformer-XL standard |           1x           |  23.23  |    24.16    |
> | with SVD (32)                      |        12.79x       |  36.49  |    37.86    |
> | with Distilled Emb (32)      |        12.79x       |  34.35  |    35.51    |
> | with Distilled Emb (64)      |         6.32x        |  26.81  |    27.63    |
>
> We pay a steeper price (in perplexity lower is better) for compressing the embedding layer of the transformer-xl language model however we have a lower perplexity than SVD. We will complete the comparison against other techniques on the same baseline.
>
>
> d) We ran a sensitivity analysis on the Pt-En translation. Based on the result, the experiments are not very sensitive to the value of alpha. We did not tune the alpha for our different experiments but chose the one which gave us good validation results on En-De translation. These results suggest that we can gain a little performance if we tune alpha for every dataset.
>
> Pt-En:
>
> | Alpha | BLEU |
> |     ---    |    ---    |
> | 0          | 42.50 |
> | 0.01    | 42.62 |
> | 0.1      | 42.65 |
> | 0.3      | 42.66 |
> | 0.5      | 42.72 |
> | 0.7      | 42.57 |
> | 0.9      | 42.03 |
>
>
> [1]: https://github.com/kimiyoung/transformer-xl

---

> ### Author Response · Authors · 2019-11-15
> **Running Time and justification of approach**
>
> 1) We specifically chose RELU so that the model can learn to regularize certain embedding dimensions, which is useful when dealing with a high dimensional embedding space, further, since this will lead to a reduction in reconstruction performance, we introduce the reconstruction loss and hyper-parameter ‘alpha’, to balance out regularization and reconstruction. These were mentioned in the paper but you are right that they were not highlighted well.
> 2) The reason we did not run experimental results for measuring the inference time is that the only accurate method to do it is either on edge device or in a simulated environment. Secondly, more than inference speed, running memory reduction is also important that is where the SVD based techniques (including ours) are superior, as there is no need to reconstruct the entire embedding matrix. We ran the experiment on inference speed and the results are shown below,
> Experimental Setup: We used 1 P100 GPU (12GB), and measured the time for the forward graph on the validation dataset (size 7590), with a batch size of 1024. We averaged this time for 30 runs and summarize are results below.
>
> |              Model                        | Inference Time (Sec) |
> | Distilled Embedding (ours) |           29.23                  |
> | SVD                                         |           29.63                   |
> | Structured Embedding        |           31.18                  |
> | Base Model                           |           27.92                   |
> We did not perform experiments on Group Reduce and Tensor Train, but they are likely to perform comparably to SVD and Our Method, or even slower.

---

### Official Review · AnonReviewer3 · 2019-10-30
**Official Blind Review #3**

**Rating:** 3

**Review:**

There are many ways to reduce the memory footprint and increase speed of a neural network: weight quantisation, compression, coarse-to-fine, knowledge distillation, etc. The method proposed in this work is a specific case of knowledge distillation that focuses on the discrete-input-to-first-layer and output-layer-to-discrete-output transformations, which represent a large portion of the parameters.

The authors propose to use a variant of SVD (which can be viewed as 2 linear transformation, with a middle dimension that represents an embedding), where the first transformation is linear with a ReLu, and the second is linear. By approximating the learned matrices of the model, the experiments show that using the proposed variant of SVD gives similar predictive performance compared to the original model, with a fraction of the parameters.

However, it seems that the authors could have simply replaced the input by a 2-layer NN (first a linear+ReLu, then a Linear) to obtain the same parametrisation, but they could have learned the parameters in a end-to-end fashion. It is not clear to me why using a surrogate L2 loss within the model should give better predictive performance than a fully end-to-end trained neural network. Without this comparison, I do not think the proposed experiments are conclusive enough.



**Experience Assessment:**

I have published in this field for several years.

**Review Assessment: Checking Correctness Of Derivations And Theory:**

I assessed the sensibility of the derivations and theory.

**Review Assessment: Checking Correctness Of Experiments:**

I assessed the sensibility of the experiments.

**Review Assessment: Thoroughness In Paper Reading:**

I read the paper at least twice and used my best judgement in assessing the paper.

---

> ### Author Response · Authors · 2019-11-13
> **Response to Review #3**
>
> We sincerely thank the reviewer for their comments and suggestions to improve the paper.
>
> We present the BLEU scores for the End-to-End 2 Layer NN (E2E-NN) approach and compare with SVD and our proposed solution below:
>
> | Dataset | E2E-NN |  SVD  | Proposal |
> |       ---     |      ---      |    ---   |       ---       |
> |    En-Fr  |   37.23   | 37.44 |    37.78    |
> |    En-De |   26.14   | 26.32 |    26.97    |
> |    Pt-En  |   42.27   | 42.37 |    42.62    |
>
> The end-to-end scheme does a little worse than SVD. We think the performance improvement for SVD and our proposal both is due to a better initialization during offline training. The difference between the End-to-End 2 layer NN and our proposal is that we initialize our method off-line and use the distillation term to regularize the embedding network.

---

> > ### Comment · AnonReviewer3 · 2019-11-13
> > **Explanations**
> >
> > Thanks for making the experiments, which indeed shows improvements over the end-to-end approach. I'm not sure I have the intuition why the end-2-end approach is better than the SVD. This seems to contradict most of the recent results of deep learning literature, which show that learning end-2-end is beneficial provided the right optimisation method and regularisation scheme. I think this specific point would need to be addressed with more scrutiny.

---

> > > ### Author Response · Authors · 2019-11-14
> > > **Response to concerns raised by AnonReviewer3**
> > >
> > > Thanks for the comment. For SVD and our approach we first train a machine translation model with full embedding matrix to convergence. Then we compress the embedding off-line and plug it back and fine-tune. So essentially we have 2 training rounds.
> > >
> > > For the end-to-end approach, we train to convergence with random initialization once. It is possible that if we let it train much longer it may slowly converge to the first solution. However, if our goal is to compress a pre-trained model it would be faster to compress the embedding and fine-tune rather than training end-to-end longer.
> > >
> > > Another interesting observation is in Table 4 in our paper. If we retain the pretrained model weights and initialize the embedding randomly we do worse than initializing everything randomly:
> > >
> > >
> > >                 Random Init.     Model Init. + Random emb.       Proposal
> > >
> > > En - Fr           37.23                              37.04                               37.78
> > > En - De         26.14                               26.07                               26.97
> > > Pt - En          42.27                               42.29                               42.62
> > >
> > >
> > > So fully random is slightly better than initializing the model with the pre-trained weights and the embedding randomly. Initializing both, our proposal and proposal of Shi & Yu (2018) and Chen et. al (2018), is the best. We agree that further experimentation would help us to generalize our findings better.

---

### Decision · Program_Chairs · 2019-12-19

**Decision:**

Reject

**Comment:**

This paper proposes to further distill token embeddings via what is effectively a simple autoencoder with a ReLU activation. All reviewers expressed concerns with the degree of technical contribution of this paper. As Reviewer 3 identifies, there are simple variants (e.g. end-to-end training with the factorized model) and there is no clear intuition for why the proposed method should outperform its variants as well as the other baselines (as noted by Reviewer 1). Reviewer 2 further expresses concerns about the merits of the propose approach over existing approaches, given the apparently small effect size of the improvement (let alone the possibility that the improvement may not in fact be statistically significant).